# Benefits of *Crotalaria juncea* L. as Green Manure in Fertility and Soil Microorganisms on the Peruvian Coast

**DOI:** 10.3390/microorganisms12112241

**Published:** 2024-11-06

**Authors:** Gregorio J. Arone, Roger Ocaña, Arcadio Sánchez, Pablo J. Villadas, Manuel Fernández-López

**Affiliations:** 1Facultad de Ingeniería, Universidad Nacional de Barranca (UNAB), Av. Toribio Luzuriaga 376 Urb. La Florida, Barranca 15169, Peru; rocanau182@unab.edu.pe; 2Facultad de Ciencias Agrarias, Universidad Nacional de Huancavelica (UNH), Av. Evitamiento Este S/N, Acobamba 09381, Peru; arcadio.sanchez@unh.edu.pe; 3Soil and Plant Microbiology Department, Estación Experimental del Zaidín, Consejo Superior de Investigaciones Científicas (CSIC), 18008 Granada, Spain; pvilladas@eez.csic.es (P.J.V.); manuel.fernandez@eez.csic.es (M.F.-L.)

**Keywords:** green manures, crotalaria, metabarcoding, soil microbiome, soil quality, soil fertility

## Abstract

The soils of the Barranca valley are among the best soils in Peru, but with so many years of application of agrochemicals and other agricultural practices, they are losing their productive capacity. Consequently, it was suggested to assess the impact of *Crotalaria juncea* L. as a green manure on soil fertility and the populations of bacteria and fungi present in the soil. Crotalaria was cultivated for 75 days and incorporated as green manure. After 90 days, the presence of bacteria and fungi was evaluated. Metabarcoding was employed, and the 16S rRNA and ITS2 amplicons were sequenced using the Illumina MiSeq platform. The sequences were processed using various bioinformatics tools. The results indicate that soils have a high diversity of bacteria and fungi. Likewise, in plots where the presence and action of natural biocontrol is suppressed (C0 and P0), pathogenic fungi increase their population in the fallow period (C1), while in P1, the addition of Crotalaria as a green manure promotes an increase in the population of bacteria and fungi, and at the same time it suppresses pathogenic fungi. The genera (bacterial and/or fungal) that increase due to the effect of Crotalaria are beneficial because they are involved as degraders of organic matter, promotion of plant growth and biological control of pathogens. Crotalaria is an alternative to improve soil fertility, increase the beneficial bacterial population, and reduce pathogenic fungi.

## 1. Introduction

Zero hunger is one of the sustainable development goals, and achieving this objective requires the presence of healthy soils, which we currently lack, because most of our soils where we produce our food are in the process of degradation [1,2]. In addition, the loss of the productive capacity of the soils seems easy to restore through the application of agrochemicals, but it seems that the remedy is more harmful, because applications of these inputs for decades have been disturbing the environment and life in the soil, etc., and has made families more dependent on these inputs, especially for family farming.

On the other hand, the soils of the Peruvian coast, although mostly alluvial, are poor from their genesis in their organic matter content and therefore in N. The soils of the Barranca valley are no exception, but in these conditions, various fruit trees, hard yellow corn, potatoes, paprika, strawberries, and other crops are grown, mainly using high doses of chemical fertilization, frequent pesticide applications, and intensive use of agricultural mechanization [3]. Over the years, dependence on and addiction to the use of agrochemicals show their effects in various ways in crop fields, such as compaction, salinization, reduction in the productive capacity of the soil [4,5], reduction in the population of soil macro-, meso-, and microorganisms, eutrophication of water, resistance, resurgence, and appearance of new pests and diseases, in addition to the contamination of the environment, soil, water, and even the crops themselves [6,7]. This reality is increasingly accentuated because there is little replacement of organic sources to the crop fields, because farmers in the valley have established a bad practice, the burning of stubble, the effects of which are widely documented [6,8,9,10]. In this way, farmers have generated a vicious circle to produce crops and obtain high yields in degraded and low-fertility soils, where they must necessarily apply higher and higher doses of fertilizer and pesticides. These practices of high fertilization have been generating changes in the characteristics of the soil, particularly in microbial populations, as well as changes in the content of organic matter, N, etc. [11,12,13].

With the urgent task of restoring the productive health of degraded or low-fertility soils, various experiments indicate that legumes used as cover or as green manure are the most accessible and low-cost allies [14,15]. However, it remains to be elucidated how much the incorporation of green manures such as Crotalaria (*Crotalaria* sp.) influences the population of bacteria and fungi that inhabit the soil, mainly using the independent method of cell culture, which makes it possible to quantify its presence and function through metabarcoding, massive sequencing, and bioinformatics that help to evaluate and make visible the contribution of these hidden inhabitants in the healthy production of crops [16].

The use of green manure is a practice known in the agricultural community since the time of our ancestors, as reported by the results of studies developed on this subject, which highlight the role of green manures in improving organic carbon content, porosity, the soil’s carbon-to-nitrogen (C:N) ratio, its structure, and soil quality [8,17,18,19]. In addition, legumes used as green manure can provide nitrogen through biological fixation, capture and sequester carbon, and provide a favorable environment for microbial biodiversity, with subsequent mineralization of plant nutrients [20,21]. Likewise, the addition of legume residues is advantageous not only for agricultural productivity but also for the sequestration of carbon from atmospheric CO_2_ [22,23,24]. Similarly, when legumes are utilized as green manure and mixed into the soil, their remains enhance the availability of nitrogen, phosphorus, potassium, and micronutrients for rotation cultivation due to the decrease in soil pH caused by the CO_2_ produced during the decomposition process [24,25,26,27].

Crotalaria (*Crotalaria juncea* L.) is employed as a cover crop for green manure, a fiber crop, fodder, as a nematicide, etc. It is also a crop that fixes atmospheric nitrogen through symbiosis and in turn provides abundant organic matter, which helps enhance the chemical, physical, and biological aspects of the soil [28,29,30,31,32,33].

The use of Crotalaria as a green manure in Barranca is a viable low-cost alternative; it adapts to its summer climate, presents rapid and exuberant growth, does not require additional fertilization or pest and disease control, and it also contributes to supply nutrients to the soil and promotes populations of bacteria and fungi that contribute to soil health. It also reduces the growth of fungi pathogenic to crops.

With this background, we aimed to know the effect of the use of Crotalaria, as green manure, on the microbial communities in order to try to recover the soil fertility and to control the presence of pathogens which impaired the crops yield.

## 2. Materials and Methods

### 2.1. Field Site and Sampling

The experiment was carried out between December 2022 and August 2023 in the village of Las Huertas, district of Barranca, Lima, Peru, located at coordinates 10°41′18.7″ South Latitude and 77°42′39.3″ West Longitude at 277 m a.s.l., which according to Holdridge corresponds to the life zone of Subtropical Desiccated Desert (dd-S) [34] (Figure 1).

An intensively cultivated plot was selected that uses pesticides and synthetic fertilizers for crop production. In addition, the owner practices the burning of agricultural stubble in the same plot, which has affected the productive capacity of the soil. An experiment of four treatments was installed in this plot: C0, P0, C1, and P1, as indicated in Table 1. During the experiment, two soil samples were taken at different times. The first sampling was carried out before the crotalaria (time zero: C0 and P0) were planted, while the second was carried out 90 days after its incorporation (time one: C1 and P1).

Soil samples were collected from a depth of 0 to 20 cm, with 12 replicates gathered for each treatment. All samples were placed in sterile Falcon tubes with lids, stored in a refrigerator, and then transported to the Genomics and Molecular Biology laboratory at the National University of Barranca, where they were kept at −80 °C until DNA extraction [35].

### 2.2. Analysis of the Leaf and Root Tissue of the Crotalaria

Crotalaria is a valuable legume as a green manure due to its multiple agronomic benefits. It fixes atmospheric nitrogen, enriching the soil and reducing the need for fertilizers. It controls phytopathogenic nematodes and certain harmful fungi, improving soil health. Its root system improves soil structure, prevents erosion, and increases the retention of water and organic matter. In addition, it is ideal for crop rotations, interrupting pest and disease cycles [36,37,38]. Crotalaria, therefore, is essential for sustainable agricultural practices, improving the productivity and health of the agricultural ecosystem. In this experiment, the Crotalaria crop was not fertilized; it grew with natural soil fertility and efficient biological nitrogen fixation [39,40].

The leaf and root tissues of the Crotalaria were sampled one day before the incorporation of the Crotalaria as green manure, which was sent in four replicates for foliar and root analysis to the Laboratory of Analysis of Soils, Plants, Water and Fertilizers (LASPAF) of the National Agrarian University La Molina (UNALM), Lima, Peru [41,42].

### 2.3. Soil Biogeochemical Properties

The soils of Barranca have been formed from sediments of alluvial origin, that is, with materials transported and deposited by water, especially by rivers and streams, so according to the current classification of the World Reference Base for Soil Resources (WRB) they correspond to Fluvisol soils [43]. These soils are the best soils in the valley, due to their rich mineral composition and the constant renewal of nutrients through the sediments that the rivers bring; however, for many decades it has been receiving overdoses of synthetic fertilizers and pesticides, which has been degrading the soil.

According to Bazán [44], soil samples were taken for characterization analysis. Four replicates were taken per treatment and sent for analysis to the Laboratory for Analysis of Soils, Plants, Water and Fertilizers (LASPAF) of the National Agrarian University La Molina (UNALM), Lima, Peru, where the following parameters were determined: (i) Texture (hydrometer method), (ii) pH (potentiometer, 1:1 water/soil), (iii) electrical conductivity (conductometer, aqueous extract 1:1 soil/water); (iv) free carbonates (gasovolumetric), (v) soil organic content (Walkley and Black), (vi) P (modified Olsen) and available K (ammonium acetate extraction, pH 7), (vii) CIC (ammonia acetate, pH 7), and (viii) exchangeable cations (atomic absorption). The analyses were performed according to the LASPAF protocol [45].

### 2.4. DNA Extraction and Illumina MiSeq Sequencing

250 mg of soil was used to extract soil DNA from 12 replicates per treatment, using the PowerSoil^®^ extraction kit (MO Bio labs, Carlsbad, CA, USA) and following the procedure described by the manufacturer. The extraction was carried out aseptically and the extracted DNA was stored at −20 °C. The concentration was quantified using the Qubit^®^ 3.0 fluorometer (Invitrogen, Life Technologies, Van Allen Way, Carlsbad, CA, USA) [35]. Once the quality and quantity of DNA had been verified, each sample was sequenced with the Illumina MiSeq platform at the Genomics Service of the Institute of Parasitology and Biomedicine “López Neyra” (CSIC; Granada, Spain) [46], following Illumina’s recommended protocols for a 2 × 275 strategy. The prokaryotic libraries were constructed amplifying the hypervariable regions V3-V4 of the 16S rRNA gene using primers Pro341F (5′-CCTACGGGNBGCASCAG-3′) and Pro805R (5′-GACTACNVGGGTATCTAATCC-3′) [47]. Fungal libraries were constructed amplifying the ITS2 region using primers ITS4 (5′-TCCTCCGCTTATTGATATGC-3′) [48] and fITS7 (5′-GTGARTCATCGAATCTTTG-3′) [49].

Three samples of the mock community ZymoBIOMICS^TM^ Microbial Community DNA Standard, (ZYMO Research; Irvine, CA, USA) were included in each sequencing run as a quality sequencing control.

### 2.5. Processing of 16S/ITS Sequences and Taxonomic Attribution

Briefly, data from metabarcoding studies were processed using R version 4.4.3 [50,51]. Amplicon Sequence Variants (ASVs) were inferred using DADA2 pipeline [52]. Trimming for prokaryotic libraries was performed with Figaro (https://github.com/Zymo-Research/figaro#figaro, accessed on 24 May 2024). For trimming, the function filterAndTrim was used. Sequences less than 50 nucleotides in length or with ambiguities were removed. Maximum Expected Errors (maxEE) for forward and reverse reads was set to 2 and 3 for prokaryotic and 3 and 4 for fungal datasets. For 16S rRNA reads, the length of the forward reads was limited to 269 nucleotides and the reverse reads to 215. Primers were removed using the Cutadapt tool [53]. After merging sequences and removing chimeras in the prokaryotic dataset, reads smaller than 401 and larger than 429 nucleotides were discarded. Finally, the classification of bacterial and fungal ASVs was achieved using the Silva 138.1 database [54] and UNITE v.7.2 dynamic database [55,56], respectively. ASVs that were classified as mitochondria, chloroplast, and unknown sequences and eukaryotic ASVs not classified as fungi at the kingdom level were removed.

### 2.6. Statistical Data Processing

In the Past4.17.exe program (PAleontological STatistics), a widely used free software for statistical analysis and ecological diversity in paleontology and biology [57,58,59], once a DataFrame was obtained, rarefaction curves were examined to understand how sampling depth affects microbial diversity calculations, and alpha and beta diversity were also determined. This software was also used for multivariate principal coordinate analysis (PCoA) with the Bray–Curtis dissimilarity index to visualize the differences in community composition between groups at 95% ellipses, for the populations of bacteria and fungi, and for the behavior of the studied soil parameters using characterization analysis. On the other hand, the various statistical comparisons of the diversity indices, the abundance of genera and phyla of bacteria and fungi, as well as the parameters evaluated in the soil characterization analysis, were analyzed using R version 4.4.3, in RStudio version 2024.04.2+764, and the library agricolae [60] at an alpha = 0.05 [50]. Finally, the construction of the Venn diagrams was carried out using Venny 2.1, an interactive online access tool. A list of genera for each soil under study (C0, P0, C1 and P1) was used, which allowed visualizing the shared and exclusive genera for each type of soil [46].

## 3. Results

### 3.1. Nutrient Supply of Crotalaria juncea *L*. as Green Manure and Changes in Soil Fertility

The leaf biomass of Crotalaria is notable for being a plentiful source of nutrients, with relatively high concentrations of nitrogen (2.54%), phosphorus (0.28%), potassium (1.33%), calcium (1.70%), magnesium (0.28%), sulfur (0.23%), zinc (21.13 ppm), copper (8.88 ppm), manganese (85.20 ppm), iron (152.33 ppm), and boron (47.91 ppm). Likewise, the root biomass of Crotalaria has considerable nutrient content, although at lower levels compared to leaf biomass. The main nutrients present in the root biomass include nitrogen (1.09%), phosphorus (0.22%), potassium (1.09%), calcium (0.46%), magnesium (0.21%), sulfur (0.22%), zinc (40.38 ppm), copper (13.25 ppm), manganese (60.88 ppm), iron (1292.50 ppm), and boron (16.83 ppm). These values show that Crotalaria used as green manure has the potential to be a significant source of nutrients to help restore soil fertility (Table 2).

The soils evaluated are soils of neutral pH, do not present salt problems, are poor in organic matter, and due to the application of phosphate and potassium fertilizers for decades the soil has a high content of P and K in the soil. Additionally, the soils exhibit low cation exchange capacity (Table 3).

The values of the parameters of the soil characterization analysis in C0 and P0, carried out at the beginning of the experiment, are presented in Table 3. On the other hand, C1 and P1 were evaluated 90 days after the incorporation of the green manure. The results of C1 and P1 were compared with each other using a *t*-test (*α* = 0.05). The soil that received the incorporation of the leaf and root biomass of the Crotalaria as green manure, 75 days after its emergence (P1), did not present statistical differences for some parameters such as P, K, Ca, Mg, Na, EC, and CEC, with respect to the plot where it was not incorporated (C1). However, there were statistical differences for pH and soil organic matter (OM) content (Table 3 and Figure 2).

Likewise, using the values of the characterization analysis of C0, C1, P0, and P1, a multivariate analysis of principal coordinates (PCoA) was performed, which has made it possible to visualize the variations in the composition of the soil. In this multivariate analysis, P1, which represents the plot P0 after the incorporation of *Crotalaria juncea* as green manure, shows a lower deviation of P0 in the PCoA space. This suggests that *Crotalaria juncea* has had a conservative effect on soil characteristics, keeping it closer to its initial state. Instead, the fact that C1 has shifted significantly from C0 in coordinate space suggests that there is a considerable change in soil characteristics when green manure is not applied. This distancing is possibly due to the processes of natural recovery of the soil, which involves a series of biological, chemical, and physical mechanisms that are natural over time (Figure 2).

### 3.2. Sequencing and Reads Quality Control

In total, 48 input libraries were subjected to ITS2 sequencing, resulting in 4,352,846 raw reads. In addition, 16S rRNA amplicon sequencing of the same number of input libraries yielded a total of 3,964,442 raw reads. The rarefaction curves based on the comparison of ASV abundance, and the number of sequences analyzed tended to reach a saturation plateau, demonstrating that the analyses were representative of the communities under investigation.

The rarefaction curves displayed in the figures illustrate the correlation between the number of sequences and the number of taxa (S) for bacteria (a) and fungi (b) under different soil conditions. Likewise, most of the curves reach a plateau, indicating that most of the bacterial and fungal diversity present in the evaluated soils has been captured (Figure 3).

### 3.3. Diversity of Bacteria and Fungi

ASVs from 16S rRNA and ITS2 were classified using a 99% sequence similarity threshold in comparison to the Silva 138.1 database and the UNITE database, respectively. Figure 4 and Figure 5 show a graphic of taxonomic compositions for the four groups evaluated. The taxonomic composition shows that the group of bacteria is dominated by the phyla Actinobacteriota, Proteobacteria, and Chloroflexi, and fungal groups by the phyla Ascomycota and Mortierellomycota.

### 3.4. The Alpha and β-Diversity

C1 and P1 soils have a higher microbial diversity according to both indices (Simpson and Shannon) compared to C0 and P0 soils. Likewise, soils C1 and P1 do not present statistical differences, suggesting that the treatments applied in C1 (natural restoration) and P1 (microbial population promoted by green manure) have promoted greater equity and richness of bacterial species, but with a diverse profile (Figure 4 and Figure 6). Likewise, the application of green manure seems to have impacted the structure of the community, reducing uniformity and dominance in P1, by influencing some bacterial genera (Figure 6).

On the other hand, for Fungi, the C1 soil reached the highest levels for the Simpson, Shannon, and Evenness indices, while P1 has behavior similar to C0 and P0. According to the values achieved in the Shannon index, fungi have less diversity than the bacterial population. On the other hand, according to the Simpson index, C1 has a high value, suggesting that one or a few species are much more abundant. In this regard, C1 has 30,584 sequences, of which 13,907 sequences, representing 45.47%, correspond to the sum of *Cladosporium* (9384 sequences) and *Fusarium* (4523 sequences) sequences (Figure 5 and Figure 7).

Figure 8 presents two principal component analysis (PCoA) graphs, which visualize the grouping of different treatments (P0, P1, C0, C1) based on the composition of the soil microbial community. In both graphs, the P1 and C1 treatments are grouped into different clusters, which indicates that these treatments have a significant and differentiating effect on the population of bacteria and fungi present in the soil. This finding suggests that both green manure treatment (P1) and soil resting period (C1) affect the soil microbial community differently.

On the other hand, P0 and C0 treatments are grouped in a separate cluster, showing similar characteristics and less differentiation compared to P1 and C1. This suggests that the soils of these treatments have a more homogeneous microbial composition or less influenced by the specific factors evaluated in P1 and C1. In summary, the incorporation of green manure (P1) and the soil rest period (C1) distinctively influence microbial diversity, clearly differentiating between P0 and C0 (Figure 8).

### 3.5. Incorporation of Crotalaria as Green Manure Restricts Some Pathogenic Fungi in Soil

Since the advent of the green revolution, the soils of the valleys of the Peruvian coast have been cultivated every agricultural season using mainly synthetic fertilizers and pesticides. Throughout these years, the continuous application of various products for the control of weeds, insects, nematodes, and fungi has reduced the beneficial biota of the soil. Thus, in soil without biocontrol, pathogenic fungi quickly recover their population. On the other hand, it is appreciated that the introduction of Crotalaria as a green manure acts as a suppressant for various plant pathogenic fungi, among them *Curvularia*, *Cladosporium*, *Alternaria*, *Aspergillus*, *Fusarium*, *Stemphylium*, and *Lasidioplodia*. These genera include species that are important pathogens in crops, causing diseases such as wilt, root rot, and leaf spots, and leading to significant yield loss in crops in the region (Figure 9). Thus, the introduction of Crotalaria as a green manure has a positive effect by restricting certain pathogenic fungi in the soil. This is because *Crotalaria juncea* and other species of the genus Crotalaria produce bioactive compounds that may have nematicidal and fungicidal properties, which contribute to reducing the incidence of pathogens in the soil and improve soil health.

### 3.6. The Incorporation of Crotalaria as Green Manure Promotes Beneficial Soil Fungi

Utilizing Crotalaria as green manure not only enhances soil fertility but also fosters a conducive environment for the growth of beneficial fungi. The clear presence of *Mortierella*, *Humicola*, *Pyxidiophora,* and *Chaetomium* in the soil promoted by green manure is important, because these genera of fungi play crucial roles in soil health and productivity, whether through the decomposition of organic matter, the promotion of plant growth, or the biological control of pathogens. Integrating practices that encourage the presence and activity of these fungi can result in more sustainable and productive agricultural systems. Therefore, incorporating Crotalaria is an effective strategy to promote microbial biodiversity and improve soil resilience to pathogens and adverse conditions (Figure 10).

### 3.7. Incorporation of Crotalaria as a Green Manure Promotes Beneficial Soil Bacteria

Although the soils show high content of P and K, and possibly also of N due to overfertilization with this element, they are poor in the most important constituent, in organic matter, so the application of any organic source will always motivate the multiplication of soil organisms, as in this experiment, the incorporation of Crotalaria as green manure promotes various bacterial genera, among them *Streptomyces*, *Microvirga*, *Gemmata*, *Agromyces*, *Ensifer*, *Glycomyces*, *Mycoplana*, *Microbacterium*, *Promicromonospora*, *Stenotrophobacter*, *Nonomuraea*, *Devosia*, *Kribbella*, *Limibaculum*, *Singulisphaera*, *Pseudaminobacter*, *Mesorhizobium*, *Nocardia*, and *Sphingomonas,* and they present statistical differences (Figure 11). These genera have various activities; for example, *Streptomyces* is known to produce natural antibiotics, controlling soil pathogens. *Microvirga*, *Ensifer,* and *Mesorhizobium* act as symbionts of legumes and fix nitrogen. Furthermore, *Agromyces* and *Mycoplana* promote plant growth through the solubilization of nutrients such as phosphorus. Likewise, *Glycomyces* and *Promicromonospora*, *Kribbella,* and *Limibaculum* participate in the decomposition of organic matter and produce antifungal compounds, which contribute to biological control. *Gemmata* and *Singulisphaera* facilitate the mineralization of nutrients, making them available to plants. *Devosia* and *Pseudaminobacter* help in the degradation of organic compounds and pollutants, improving soil quality (Figure 11).

### 3.8. Venn Diagrams of Soil Bacteria and Fungi Promoted by Green Manure

The soils of La Huerta, Barranca, are represented by 307 unique bacterial genera, which are shared in soils C0 (161), C1 (206), P0 (183), and P1 (200). On the other hand, for fungi they are represented by 145 unique genera, shared in C0 with 108, C1, 125, P0, 107, and P1, 88 genera. The Venn diagrams show that 111 bacterial genera (36.2%) are common to all soil conditions, whereas C1 and P1 have 41 (13.4%) and 35 (11.4%), respectively, as their own. On the other hand, 66 genera (45.2%) of fungi are common to all the soils evaluated, whereas C1 and P1 promote 13 (8.9%) and five (3.4%) as their own, respectively (Figure 12).

P1-specific bacteria include among others *Saccharothrix, Phytomonospora,* and *Actinocorallia* which belong to the actinomycetes and have various beneficial functions in the soil. C1 also has its own genera, such as *Pseudonocardia, Saccharopolyspora, Prauserella,* and *Actinophytocola*, which belong to the actinomycetes and are considered to perform various functions in the soil, including protecting plants against pathogens through the production of antibiotics and improving soil fertility through the decomposition of organic matter. Benefits that in the time of the Incas were taken advantage of, because by resting the soils the fertility of the soil was recovered and when they returned to cultivate the plots the soil did not require external inputs, and production was achieved with the natural fertility of the soil accumulated over several years (5 to 15 years).

## 4. Discussion

Plants belonging to the Crotalaria genus are characterized by their fast vegetative growth, substantial biomass production, efficient extraction of nutrients from the soil, and exceptional adaptability to low soil fertility conditions [17,61], and depending on the conditions they can provide around 7.9 t/ha of *Crotalaria juncea* L. biomass, approximately 223 kg/ha of nitrogen, 27 kg/ha of phosphorus, and 247 kg/ha of potassium to the soil [62], similar to the results achieved in this experiment (Table 2).

It is also important to note that Crotalaria as green manure not only provides nutrients to the soil, but also offers other benefits such as fixing nitrogen, its nematicide action, controlling weeds, improving fertility, and promoting soil structure [39,63,64,65,66,67] (Table 3). It also highlights the ability to suppress pathogenic fungi from the soil (Figure 5) and other benefits, which make it an attractive option for farmers, so that its incorporation into agricultural systems promotes soil health and contributes to reducing farmers’ addiction to external inputs, such as synthetic nitrogen fertilizers and pesticides [68,69,70,71].

The high P content in these soils can be attributed to the synthetic fertilizers that the farmers have used for many decades to fertilize their crops (Table 3). In addition, its low efficiency, which ranges from 15 to 30% [72], means that farmers use more than the doses that the plants need. Five weeks after its application to the soil, about 60% of the P is insolubilized; in this way, the precipitation of phosphorus is the predominant mechanism that reduces its availability [73]. Therefore, the soils evaluated are soils with high P content and are statistically similar, where the contribution of P by the green manure (26.61 kg per ha) is quite low (Table 2). Similar behavior is observed in terms of the high K content in the soil, possibly due to the fixation of potassium from many years of application [74,75,76]. Green manure contributes 116.70 kg of K in its leaf biomass and 10.50 kg in its root biomass, and in total it contributes 127.20 kg of K per ha, This contribution is lower than those reported by [62], who indicate a contribution of 364.25 kg of K per ha, and also clarify that the largest reserve of K is found in the stems of green manure (Table 2 and Table 3 and Figure 2).

The application of green manures helps to increase the pH of the soil due to the various processes that the organic source undergoes, such as the microbial decomposition of carbohydrates, ammonium mineralization, the high content of cations that form bases of vegetable residues such as the contribution of Ca in the biomass of the Crotalaria and the amount of vegetable residue, etc. (Table 2; Figure 2b). Likewise, the increase in pH in the soil is due to the decarboxylation of organic ions and the bonding of hydrogen ions (H^+^) with organic anions and other chemically negatively charged functional groups, and this process varies according to the type of residue and the initial pH of the soil [77].

Organic matter together with other factors conditions the activity of soil macro- and microorganisms. The more plant biomass is contributed to the soil, the greater the quantity and variety of microorganisms that will be favored because organic carbon content is a key factor for soil health and plays an important role in mitigating the effects of climate change [78,79]. On the other hand, soils poor in organic matter, such as Barranca (Table 2 and Table 3), require urgent replacement of organic matter, so the inclusion of green manure crops that provide high-energy organic matter favors the increase in microbial biomass and other functions [8,19,80].

Once incorporated into the soil, green manure promotes various bacterial genera, which when compared with other soils present statistical differences. We have *Streptomyces*, *Nocardia*, *Microvirga*, *Gemmata*, *Agromyces* (Figure 11), *Mycobacterium*, *Promicromonospora*, *Devosia*, *Kribbella*, *Ensifer*, *Glycomyces*, *Mycoplana*, *Stenotrophobacter*, *Nonomuraea*, *Mesorhizobium*, *Pseudaminobacter*, *Singulisphaera*, and *Limibaculum*, of which the genera *Streptomyces*, *Microvirga*, *Ensifer (Sinorhizobium)*, *Devosia,* and *Mesorhizobium* are widely known for their activity as Plant Growth-Promoting Rhizobacteria (PGPR) [81,82,83,84]. Likewise, in these environments where various pesticides have been applied for many years and continue to be applied, the presence of the genera *Streptomyces*, *Microvirga*, *Mycobacterium*, *Devosia*, and *Nocardia* is good, because they have species that are involved in bioremediation processes, especially in the degradation of hydrocarbons and other organic pollutants; likewise, they have enzymes to degrade soil organic matter [84,85,86,87].

Currently, in the agricultural fields of Barranca, the presence of pathogenic fungi reduces crop growth and yield and, in severe cases, destroys crops, leading to economic losses for families [88,89]. As a result, farmers spend large sums of money protecting their crops with constant applications of fungicides and other chemicals to control pathogenic fungi, which increases production costs [90]. On the other hand, Barranca is considered the best paprika production center, but it presents a problem in production due to the presence of aflatoxins. These mycotoxins are produced by various fungi, *Fusarium*, *Alternaria*, *Penicillium*, and *Aspergillus*, which contaminate crops from the field [91,92,93].

Crotalaria, when used as green manure, can help suppress the development of soil pathogenic fungi. This action is possible because Crotalaria contains sulfur compounds such as glucosinolates that, when incorporated into the soil, release isothiocyanates with biofumigant properties, which in addition to other secondary metabolites, such as pyrrolizidine alkaloids, flavonoids, and phenolic compounds, have inhibitory effects on various pathogenic fungi in the soil, such as *Fusarium*, *Rhizoctonia,* and *Pythium* [68,70,71].

The soils evaluated are soils poor in organic matter. On the other hand, they have a high content of P and K, and possibly also of N because high doses of nitrogen fertilization are applied. Likewise, the application of synthetic fertilizers, especially NPK, increases the relative abundance of fungal phytopathogens such as *Alternaria* and *Fusarium*. In other experiments, the pathogen burden of the community increases with fertilization intensity, because fertilization promotes the abundance of the pathogen and increases the severity of crop disease [94,95]. Apart from synthetic fertilizers, pesticides, and some agricultural practices such as stubble burning, they contribute to reducing the population of beneficial microorganisms in the soil, so there is less resistance to the increase in pathogens in crop fields [96,97] (Figure 9). In the soils evaluated in C1, plant pathogenic fungi are represented with a high number of sequences, mainly by *Cladosporium* (24.23%), *Fusarium* (11.68%), *Alternaria* (3.85%), and *Rhizoctonia* (0.96%). On the other hand, in P1 their presence is reduced for *Cladosporium* (10.33%), *Fusarium* (8.18%), *Alternaria* (0.77%), and *Rhizoctonia* (0.36%). Likewise, fungi that degrade soil organic matter, such as *Mortierella* and *Humicola*, in C1 are represented by 3.98% and 2.95%, respectively. On the other hand, in P1, *Mortierella* represents 40.06% and *Humicola* 13.99% (Figure 9 and Figure 10). This indicates that the restoration of soil biota, in a field conducted under the system of intensive agriculture and high inputs, is completely different from the restoration of soil biota conducted under the organic farming system [35].

The introduction of Crotalaria and its incorporation as green manure is a cheap and affordable source to supply organic matter to the farms. This process benefits the presence of *Mortierella*, a genus that thrives as a saprophyte of organic matter, certain strains of this genus are categorized as plant growth-promoting fungi (PGPF) and enhance the availability of phosphorus in agricultural soils, produce siderophores, generate phytohormones, and contribute positively to plant growth [83,98,99]. *Chaetomium* is also recognized for its biocontrol potential and its ability to suppress the growth of fungi and bacteria through competition. Some species act as endophytes [100,101,102]. On the other hand, *Humicola* is a saprophytic fungus important for the mineralization of organic carbon and nitrogen. This species uses various sources of carbon, from the simplest sugars to more complex molecules such as organic acids, disaccharides, starch, pectin, cellulose, fats, and lignin. Due to its thin and extensive hyphae, it is able to explore different sources of organic carbon, while at the same time it encourages the aggregation of particles and improves the structure of the soil [103,104,105,106]. On the other hand, *Streptomyces* is a Gram-positive soil bacterium that synthesizes a range of advantageous secondary metabolites, including antibiotics and enzymes, which safeguard plants against pathogens and foster their growth [107,108,109,110].

## 5. Conclusions

Crotalaria adapts very well in Barranca. It has a fast and exuberant growth, and it does not need extra fertilization, nor control of pests and diseases, so it is an alternative to improve soil fertility. It provides 9723 kg of dry biomass and 233 kg of N per ha and growing season. It also reduces plant pathogenic fungi. In P1, the presence of Crotalaria reduced the presence of *Cladosporium* (10.33%—3892 ASV), *Fusarium* (8.18%—3082 ASV), *Alternaria* (0.77%—292 ASV), and *Rhizoctonia* (0.36%—137 ASV); on the other hand, in C1, a fallow plot, a greater presence of *Cladosporium* (24.23%—9384 ASV), *Fusarium* (11.68%—4523 ASV), *Alternaria* (3.85%—1492 ASV), and *Rhizoctonia* (0.96%—371ASV) was found. Likewise, when the presence and action of natural biocontrol in the soil is suppressed (C0 and P0), and pathogenic fungi are controlled exclusively with chemicals (C0 and P0), when the soil goes fallow, the population of pathogenic fungi is the population with the greatest presence (C1).

The soils evaluated present a high diversity of bacteria and fungi as indicated by the Shannon index. Likewise, when Crotalaria is added as a green manure, it promotes various genera such as *Streptomyces*, *Microvirga*, *Ensifer (Sinorhizobium)*, *Devosia,* and *Mesorhizobium*, which are favored. These genera are known to have species that act as PGPRs. Likewise, green manure promotes a population of fungi involved in the use and degradation of organic sources, such as *Mortierella* and *Humicola*.

## Figures and Tables

**Figure 1 microorganisms-12-02241-f001:**
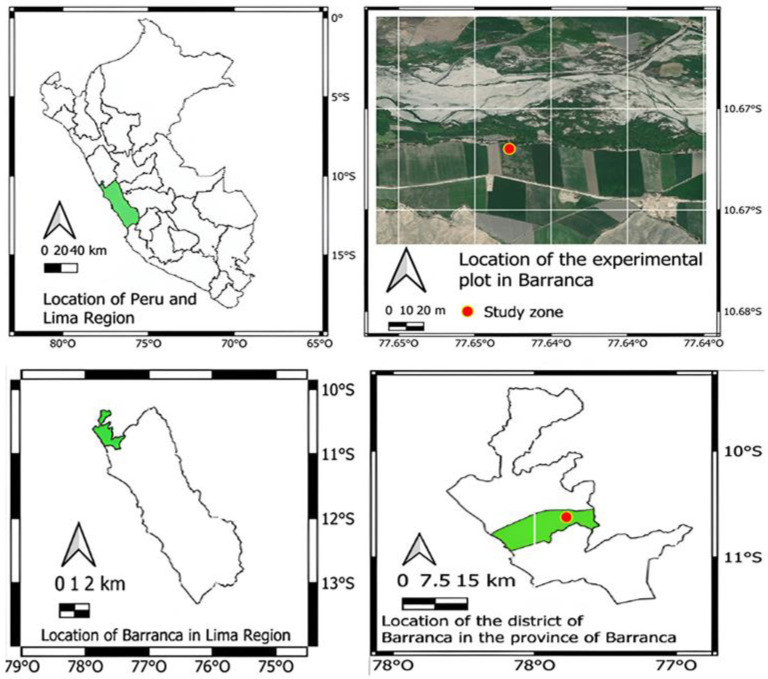
Location of the experimental plot (red dot) in La Huerta, Barranca, Lima, Peru.

**Figure 2 microorganisms-12-02241-f002:**
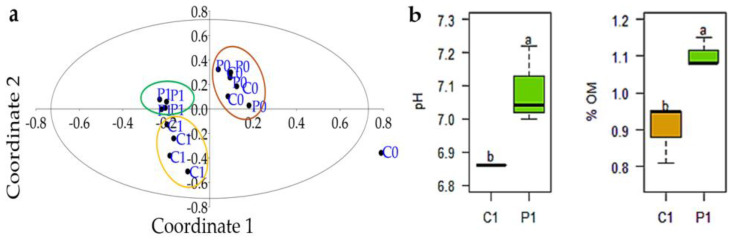
(**a**) Multivariate analysis of principal coordinates (PCoA) using the values of the characterization analysis of C0, C1, P0, and P1. (**b**) Soil pH and organic matter content. Significant differences between treatments for pH and soil organic matter are indicated by the letters “a” and “b”.

**Figure 3 microorganisms-12-02241-f003:**
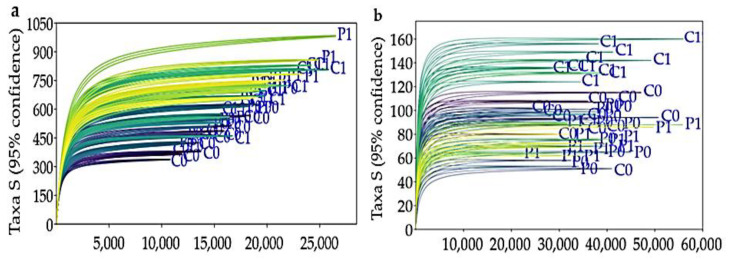
Rarefaction curves of bacteria (**a**) and fungi (**b**) of soils evaluated.

**Figure 4 microorganisms-12-02241-f004:**
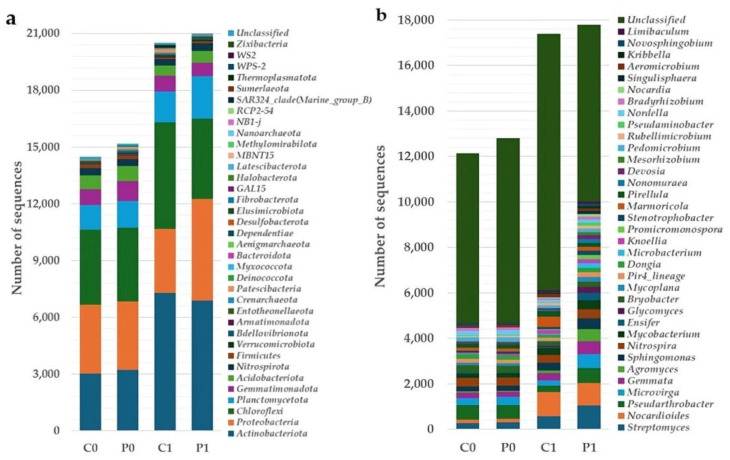
Composition of bacterial communities by phylum (**a**) and genus (**b**) in soil C0, P0, C1, and P1 expressed as absolute number of sequences.

**Figure 5 microorganisms-12-02241-f005:**
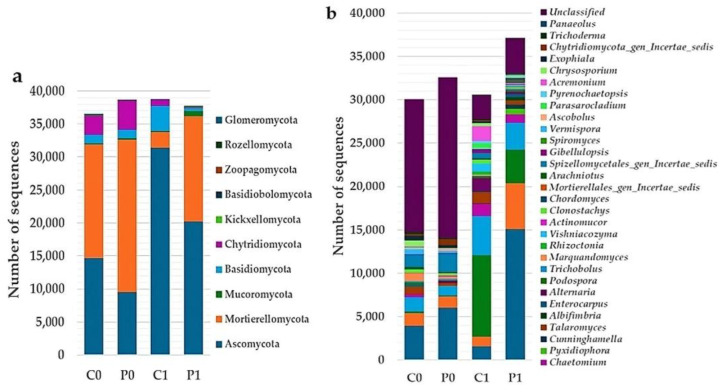
Composition of fungal communities by phylum (**a**) and genus (**b**) in soil C0, P0, C1, and P1 expressed as absolute number of sequences.

**Figure 6 microorganisms-12-02241-f006:**
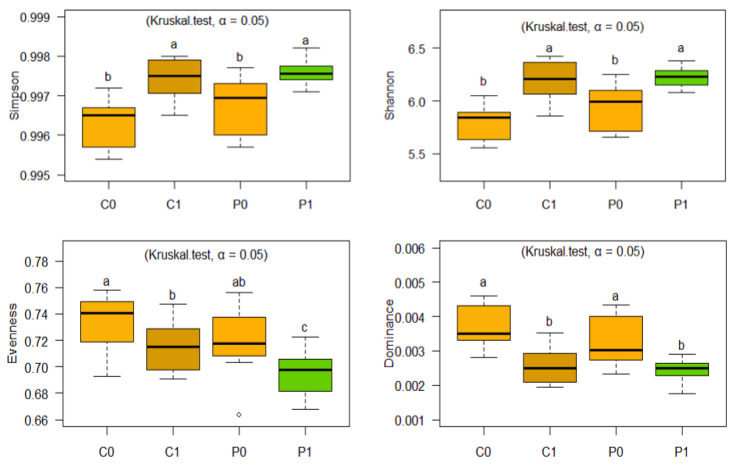
Simpson’s Index, Shannon, Dominance, and Evenness of Soil Bacteria. The letters “a”, “ab”, “b”, “c” indicate significant differences between treatments. Same-letter treatments are not significantly different from one another. On the other hand, intermediate combinations such as “ab”, show that a treatment shares similarities with both groups.

**Figure 7 microorganisms-12-02241-f007:**
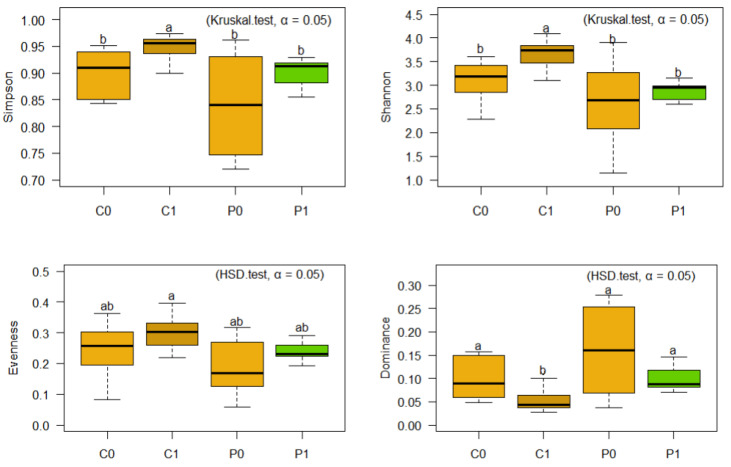
Simpson’s Index, Shannon, Dominance, and Evenness of Soil Fungi. The letters “a”, “ab”, “b”, indicate significant differences between treatments. Same-letter treatments are not significantly different from one another. On the other hand, intermediate combinations such as “ab”, show that a treatment shares similarities with both groups.

**Figure 8 microorganisms-12-02241-f008:**
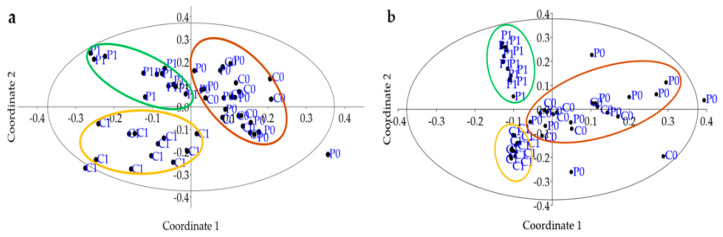
PCoA of soil bacteria (**a**) and fungi (**b**) promoted by green manure in Barranca.

**Figure 9 microorganisms-12-02241-f009:**
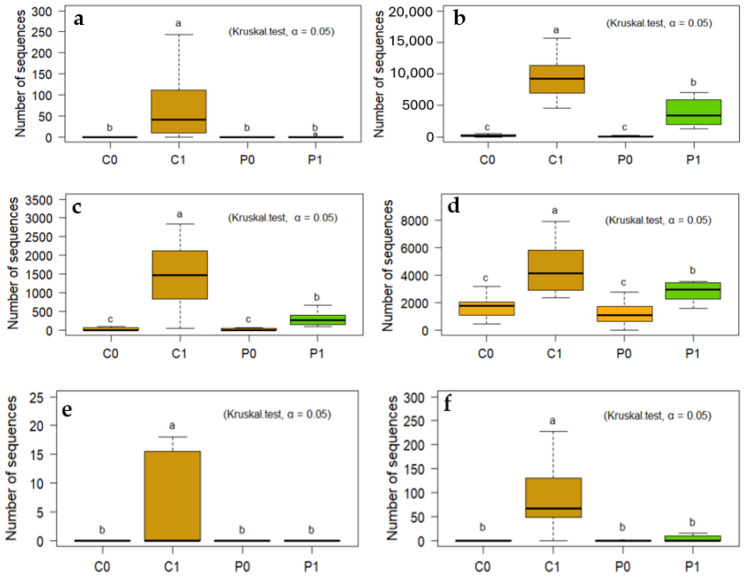
Crotalaria as a green manure acts as a suppressant of pathogenic fungi of (**a**) *Curvularia*, (**b**) *Cladosporium*, (**c**) *Alternaria*, (**d**) *Fusarium*, (**e**) *Lasidioplodia*, and (**f**) *Stemphylium.* The letters “a”, b”, “c”, indicate significant differences between treatments. Same-letter treatments are not significantly different from one another.

**Figure 10 microorganisms-12-02241-f010:**
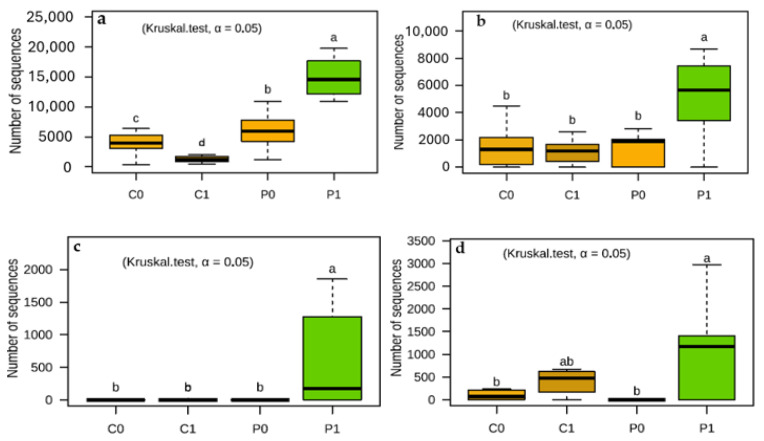
The use of Crotalaria as green manure promotes a population of beneficial fungi such as (**a**) *Mortierella*, (**b**) *Humicola*, (**c**) *Pyxidiophora*, and (**d**) *Chaetomium*. The letters “a”, “ab”, “b”, “c”, “d” indicate significant differences between treatments. Same-letter treatments are not significantly different from one another. On the other hand, intermediate combinations such as “ab”, show that a treatment shares similarities with both groups.

**Figure 11 microorganisms-12-02241-f011:**
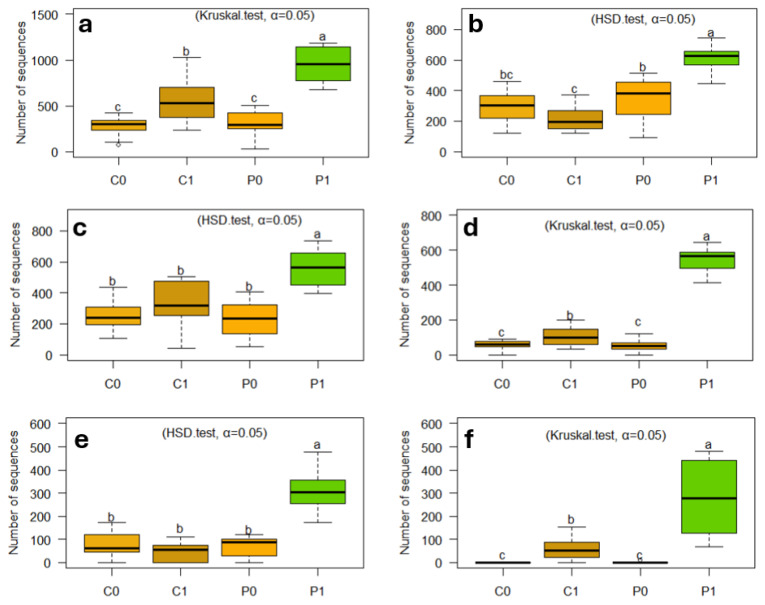
The use of green manure Crotalaria promotes genera of beneficial soil bacteria such as (**a**) *Streptomyces*, (**b**) *Microvirga*, (**c**) *Gemmata*, (**d**) *Agromyces*, (**e**) *Ensifer*, and (**f**) *Glycomyces.* The letters “a”, “b”, “bc”, “c” indicate significant differences between treatments. Same-letter treatments are not significantly different from one another. On the other hand, intermediate combinations such as “bc”, show that a treatment shares similarities with both groups.

**Figure 12 microorganisms-12-02241-f012:**
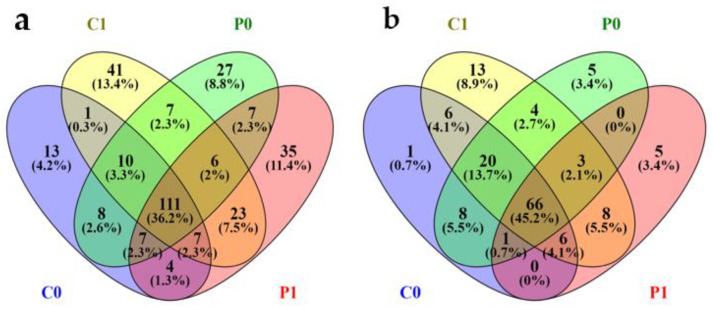
Venn diagrams of bacteria (**a**) and fungi (**b**) of C0, C1, P0, and P1.

**Table 1 microorganisms-12-02241-t001:** Treatments under study.

Time Zero	Time 1	Plots with and without Crotalaria Sowing for Use as Green Manure
C0: Initial control soil evaluated before sowing of the plot without cultivation.	C1: Soil that has remained untilled since the time of C0 and evaluated at the same time as P1.	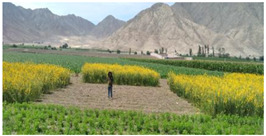
P0: Initial control soil for the plot with Crotalaria	P1: Plot where the Crotalaria is sown and incorporated 75 days after its sowing and evaluated 90 days after its incorporation.	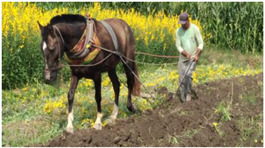

**Table 2 microorganisms-12-02241-t002:** Nutrient content of *Crotalaria juncea* L. as green manure at 75 days after its emergence in La Huerta, Barranca, Lima.

Element	Crotalaria Leaf Biomass (LB)			Crotalaria Root Biomass (RB)			Total Input (LB + RB) (kg ha^−1^)
	Nutrient Content (g kg^−1^)	Dry Matter (t ha^−1^)	Nutrient Supply (kg ha^−1^)	Nutrient Content (g kg^−1^)	Dry Matter (t ha^−1^)	Nutrient Supply (kg ha^−1^)	
N	25.40	8.767	222.70	10.90	0.968	10.55	233.23
P	2.80	8.767	24.50	2.20	0.968	2.13	26.68
K	13.30	8.767	116.60	10.90	0.968	10.55	127.15
Ca	17.00	8.767	149.00	4.60	0.968	4.45	153.49
Mg	2.80	8.767	24.50	2.10	0.968	2.03	26.58
S	2.30	8.767	20.20	2.20	0.968	2.13	22.29
Na	0.100	8.767	0.90	1.20	0.968	1.16	2.04
Zn	0.021	8.767	0.20	0.040	0.968	0.04	0.22
Cu	0.009	8.767	0.10	0.013	0.968	0.01	0.09
Mn	0.085	8.767	0.70	0.061	0.968	0.06	0.80
Fe	0.152	8.767	1.30	1.293	0.968	1.25	2.58
B *	0.048	8.767	0.40	0.017	0.968	0.02	0.44

* Macro- and micronutrients are reported in their elemental form.

**Table 3 microorganisms-12-02241-t003:** Soil characterization analysis.

Treatment	pH	C.E	OM	P	K	CEC	Ca^+2^	Mg^+2^	K^+1^	Na^+1^	Sum of Bases
		(dS/m)	(g kg^−1^)	(mg kg^−1^)		(mEq/100 g soil)					
C0	6.91	0.52	9.8	34.11	328.25	11.00	8.15	1.78	0.64	0.18	10.75
P0	7.09	0.65	10.7	31.54	268.25	11.04	8.54	1.88	0.56	0.07	11.04
C1 *	6.86 ^b^	1.12 ^a^	9 ^b^	32.25 ^a^	267.67 ^a^	10.24 ^a^	7.43 ^a^	1.77 ^a^	0.57 ^a^	0.14 ^a^	9.91
P1	7.09 ^a^	0.99 ^a^	11 ^a^	32.16 ^a^	269.00 ^a^	10.83 ^a^	8.25 ^a^	1.93 ^a^	0.58 ^a^	0.05 ^a^	10.81

* In the table provided, the significant differences between the C1 and P1 treatments are indicated by the letters “a” and “b” next to the values in the corresponding columns, where the means that do not share a letter are significantly different.

## Data Availability

Raw stream reads (SRA accession numbers SAMN43364324/ SAMN43364515) are available in the NCBI GenBank database under BioProject the access number PRJNA1152496 (Accessible from 1 October 2024).

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
