# Peer review of "Benefits of Crotalaria juncea L. as Green Manure in Fertility and Soil Microorganisms on the Peruvian Coast"

_microorganisms, 2024, doi:10.3390/microorganisms12112241_

Round 1
Reviewer 1 Report
Comments and Suggestions for Authors
In the article, the authors studied the influence of Crotalaria juncea L. on the composition of the microbial and fungal community of soils. The authors used several methods to determine the physicochemical properties of the studied soils, as well as amplicon sequencing to determine the structure of microbial and fungal communities. As a result of the study, it was found that Crotalaria juncea L. affects the composition of microbial and fungal communities (but in what way?).
The manuscript is written averagely, there are several questions and comments:
1. Was crotalaria grown in a different soil area? After crotalaria was planted in the soil, what crop was grown? Were the crops the same in the studied areas C and P?
2. Quite a lot of text from the ‘Discussion’ section should be moved to the corresponding ‘Results’ sections, since it only describes the obtained results, which is sorely lacking in the ‘Results’ section.
3. The quality of many figures is very different.
4. The abstract does not present the research data, it should be rewritten.
5. Several keywords should be added. 6. The purpose of the study should be added at the end of the introduction.
7. Section 2 should also provide the packages/programs used to perform the statistical analysis of the data obtained.
8. Line 102 - Table 1 and the abstract indicate that P1 and C1 were assessed 90 days after incorporation. How much time has passed since incorporation? How many days later were the samples taken again - 30 or 90?
9. Section 2.3 - references should be given to the methods used to determine the biogeochemical parameters of the studied soils.
10. Perhaps Table 2 should be called ‘nutrient content…’ rather than ‘nutrient contribution…’, since it provides the content of various elements in the leaves and roots of crotalaria.
11. In Table 2, the Dry Matter (kg.ha-1) columns for leaves and roots indicate the same values ​​for all elements (8766.56 and 967.89) - is this the detection limit for the device or a typo?
12. Lines 188-192 - How was the statistical analysis performed? This information should be included in a separate section on statistical processing of the data in section 2.
13. Sections 3.2 and 3.3 should be combined.
14. In figures 4 and 5, the Y-axes are labeled as ‘ASV-Phylum’ and ‘ASV-Genus’, but these are more like subtitles for the figures. The values ​​are given in absolute numbers. The correct axis titles should be given and the appropriate headings added. Also, line 235 states that the abundance of Cladosporium and Fusarium is 35%, but this is not clear from the figures with absolute numbers of reads ​​(?).
15. Also, in section 2, in the subsection on statistical processing of the data, the software used to create the illustrations should be given.
16. Line 222 - should be moved to the section on statistical data processing in section 2.
17. Lines 222-225 - Are the other indicators shown in Figure 6 unimportant? The image quality in Figure 6 is low, it should be made similar to Figure 7.
18. Information on how PCA was done and in what program should be provided in the subsection on statistical data processing in section 2.
19. Section 3.6 - The description of the data is too brief (literally 2 short sentences) for a fairly large Figure 9. It should be described in more detail.
20. Section 3.7 - Same as in section 3.6.
21. Section 3.8 - Same as in sections 3.6 and 3.7. Figure 11 shows 18(!) small diagrams on the content of various soil bacteria and there is no description. Are all 18 diagrams necessary? Then they should be described in more detail.
22. How the Venn diagrams were derived should be described in the section on statistical analysis of the data.
23. Line 287 - 145 unique fungal genera are listed. Line 286 states ‘C0 has 108, C1, 125, P0, 107 and P1, 88 genera’, which gives a total of 428. Is the number given in the text correct? The number of bacterial genera should also be checked.
24. Line 289 - ‘Crotalaria (P1) also affects the microbial community but suppresses pathogens’ - this statement does not follow from the Venn diagram.
25. Were ASVs or specific genera used to construct the Venn diagrams?
26. Line 308 - Figure 5 shows 35 fungal genera. Which of the listed genera are pathogenic and/or which fungal genera are declining in abundance should be stated here or in section 3.4.
27. Line 318 - Table 2 shows the phosphorus content in plants as a whole as 26.62 kg/ha. Probably a typo somewhere.
28. Some publications in the list of references are formatted with errors, for example, 33.
Sincerely, reviewer.
Author Response
Cordial greetings, we deeply appreciate the time and effort for the review of our work. Your feedback has been invaluable in improving the quality and clarity of the article. We have made the modifications in response to your comments which we attach.

Reviewer 2 Report
Comments and Suggestions for Authors
Comments and suggestions for Authors
Title: Benefits of Crotalaria juncea L. as Green Manure in Fertility and Soil
Microorganisms on the Peruvian Coast
The manuscript is interesting and its content falls within the publishing scope of Microorganisms journal. Major corrections and additions are required throughout the manuscript. One growing season is not enough to obtain reliable results.
Remarks:
In order to increase the usefulness of the article, Authors must refer to the following points.
§ Key words: please add the necessary keywords.
§ Materials and Methods: Please provide the name of the statistical method according to which the experiment was established. The authors write in the Introduction section that Crotalaria juncea L. is an N2-fixing plant, so why was a Rhizobium inoculum not used? Subsection 2.1. What mineral fertilization was used? Please provide the soil type according to the current WRB classification.
Subsection 2.2. - Please specify what analyses were performed, the principles of the analytical methods, and the equipment (company, city, country).
The title of subsection 2.3 should be modified, the methodology should be expanded, and references should be provided in References.
§ Results: Please use SI units - Table 2, 3 (g kg-1, mg kg-1, t ha-1). What forms of P and K are included in Table 3? In Table 3, please write down the ionic forms correctly. Please include explanations of abbreviations below the tables. Table 2 does not include data on iron and boron content. The unit of the sum of base cations must be written correctly.
§ Discussion: The number of citations for one sentence should be reduced. Lines 325-326; 332; and 336 - Citations must be spelled correctly. Line 322 - What are the organic compounds of potassium?
Specific remarks:
Please also correct the subsection numbering in the Material and Methods section.
The numbering of tables and figures should be improved.
Some References should be adapted to editorial requirements (e.g. 5, 6, 8, 20, 22, 25, 32, 33, 44, 58, 64).
Best regards
Author Response
Cordial greetings, we deeply appreciate the time and effort for the review of our work. Your feedback has been invaluable in improving the quality and clarity of the article. We have made the modifications in response to your comments which we atach.

Round 2
Reviewer 1 Report
Comments and Suggestions for Authors
Dear authors!
Thank you for correcting the comments.
Best regards, reviewer.
Reviewer 2 Report
Comments and Suggestions for Authors
Comments for Authors
Title: Benefits of Crotalaria juncea L. as Green Manure in Fertility and Soil
Microorganisms on the Peruvian Coast
Dear Authors
The authors confirmed the validity of my comments. In many responses I read that they responded to Reviewer 1. OK. I only received responses as Reviewer 2.
The manuscript has been corrected and supplemented in accordance with the comments.
Note:
In the entire manuscript, please correct the table numbers (should be: 1, 2, 3.......) and figures (should be: 1, 2,......).
Best regards